# Growth and Maturity Status of Female Soccer Players: A Narrative Review

**DOI:** 10.3390/ijerph18041448

**Published:** 2021-02-04

**Authors:** Robert M. Malina, Diogo V. Martinho, João Valente-dos-Santos, Manuel J. Coelho-e-Silva, Sławomir M. Kozieł

**Affiliations:** 1Department of Kinesiology and Health Education, University of Texas, Austin, TX 78712, USA; 2School of Public Health and Information Sciences, University of Louisville, Louisville, KY 40292, USA; 3University of Coimbra, FCDEF, CIDAF (uid/dtp/042143/2020), 3040-248 Coimbra, Portugal; dvmartinho92@hotmail.com (D.V.M.); mjcesilva@hotmail.com (M.J.C.-e.-S.); 4Faculty of Physical Education and Sport, Lusófona University, 1749-024 Lisbon, Portugal; j.valente-dos-santos@hotmail.com; 5Faculty of Sports Science and Physical Education, University of Coimbra, CIDAF (uid/dtp/042143/2020), 3040-248 Coimbra, Portugal; 6Department of Anthropology, Hirszfeld Institute of Immunology and Experimental Therapy, Polish Academy of Sciences, 53-114 Wrocław, Poland; slawomir.koziel@hirszfeld.pl

**Keywords:** height, weight, youth athletes, puberty, maturity offset, predicted adult height

## Abstract

Reported mean ages, heights and weights of female soccer players aged <19 years in 161 studies spanning the years 1992–2020 were extracted from the literature or calculated from data available to the authors; 35 studies spanning the years 1981–2020 also included an indicator of biological maturation. Heights and weights were plotted relative to U.S. reference data. Preece–Baines Model 1 was fitted to moving averages to estimate ages at peak velocity. Maturity indicators included skeletal age, pubertal status, age at menarche, percentage of predicted adult height and predicted maturity offset. Heights and weights showed negligible secular variation across the time interval. Heights were slightly above or approximated the reference medians through 14 years old and then varied between the medians and 75th percentiles through 18 years old. Weights were above the reference medians from 9 to 18 years old. Mean ages at menarche ranged from 12.7 to 13.0 years. The trend in heights and weights suggested the persistence and/or selection of taller and heavier players during adolescence, while estimated age at peak height velocity (PHV) and ages at menarche were within the range of mean ages in European and North American samples. Data for skeletal and sexual maturity status were limited; predicted maturity offset increased linearly with mean ages and heights at prediction.

## 1. Introduction

An early review of the growth and maturity status of youth athletes [1] indicated only two studies of female soccer players. One considered skeletal age (SA) among adolescent female athletes in several sports, including soccer [2], while the other described the somatotypes of 50 female athletes, including six soccer players, aged 10–15 years [3]. This is somewhat surprising given the rapid increase in the number of girls participating in soccer in both Europe and North America since the 1970s [4,5,6]. In contrast, injuries in youth soccer tournaments have been of interest and studies have noted higher injury rates among females than in males [7,8,9,10]. These studies focused on injury statistics and types of injuries; growth and maturity were not considered. 

An early review of the physiological and physical characteristics of adult female soccer players included six studies between 1986 and 1992 that reported heights and weights [11]. Subsequent reviews [12,13] noted an increase in studies of body size and composition and the functional characteristics of adult players, but one [13] included eight studies reporting age, height, weight and/or the functional characteristics of players aged ≤18 years, while the other [12] noted that “…the anthropometric profile of elite youth players has yet to be fully examined” (p. 5). In contrast, a systematic review of the reliability and utility of change of direction speed tests among female adolescent soccer players did not consider potential variation in test performances associated with chronological age (CA) per se (mean CAs of the samples ranged from 9.3 to 17.3 years) and with growth and maturity status [14]. Although variation in growth and maturation is indicated as central to the long-term development of female youth players [15], it is essential to recognize the interactions among CA, growth and maturity status, and the functional development of youth players. 

Given the increased popularity of soccer among girls and the selectivity of sport and competition for potentially talented players at relatively young ages, the purpose of this narrative review is to evaluate the growth and maturity status of female youth soccer players. Studies reporting heights and weights spanned 1992 through 2020, and, except for a study in 1981, studies reporting maturity indicators spanned 1995 through 2020.

## 2. Materials and Methods

### 2.1. Search Methods

Studies of female soccer players aged <19 years were extracted from several databases (Pubmed, Web of Knowledge and Scopus) using the following terms: “youth female soccer”, “youth soccer female”, “young soccer female”, “soccer female” and “female soccer”. In addition to age, height and weight, the search also identified studies that included an indicator of biological maturation. Initial screening was done by title and abstract, and when needed the full manuscript was consulted. If the data combined players from multiple sports or combined males and females, the manuscript was excluded. All materials were reviewed by two individuals (D.V.M. and R.M.M.), but the final decision was made by the first author. The extracted studies were complemented with papers on female athletes accumulated over the years by the first author. 

### 2.2. Age, Height and Weight

Mean CAs, heights and weights were extracted from 155 full-text articles spanning 1992 through November 2020. In addition, raw data for CAs, heights and weights were available to the first author from six studies: three theses dealing with youth players, a study of collegiate athletes that included a sample of soccer players aged 18 years and two surveys of adolescent participants in recent bio-banded soccer tournaments (see Acknowledgments). Age-group-specific means were calculated for each of the six data sets. Thus, 161 studies were deemed eligible. The studies are indicated in chronological order in Appendix A. Several studies reported the same age, height and weight data in one or more reports. The earliest report was retained for this review; the others are also indicated the Appendix A.

The 161 studies included 293 observations (age-group means) for the age, height and weight of female youth players spanning 8.2 to 18.9 years old. The majority of studies and data points were, respectively, from Europe (75 and 140) and North America (61 and 106), followed by Latin America (10 and 20), Asia-India (9 and 18), the Pacific (3 and 6) and Africa (3 and 3). The U.S., England, Spain and Canada accounted for 58% of the studies and 60% of the data points. The median sample size was 20 with a range from 6 to 4556, but sample sizes ranged from 10 to 75 for 85% of the data points. The largest sample included youth players from 230 clubs participating in a prospective study of knee injuries [16].

Prior to considering the growth status of the female players, two preliminary analyses were done. The first considered potential secular variation in body size. For the analysis of secular change, studies were grouped into three time intervals, 1992 to 2009 (27 studies, 49 data points), 2010 to 2017 (63 studies, 114 data points) and 2018 to 2020 (71 studies, 130 data points). Within each interval, composite means of the mean CAs, heights and weights were also calculated for several CA groups (<11, 11–12, 13–14, 15–16 and 17–18 years) and compared within each CA group using analyses of variance (ANOVAs). Reported means were not adjusted for sample size to avoid the risk of age-adjusted means being influenced by few studies with large samples. 

The mean heights and weights from each study in the three intervals were also plotted by CA relative to U.S. reference medians and the 25th and 75th percentiles for girls [17]. The reference was developed from nationally representative samples of American girls of European, African-American and Hispanic-American ancestry surveyed between 1963 and 1994. Heights of American girls (also boys) did not change across surveys between 1963 and 1994, but weights increased from the late 1970s to the early 1990s. As a result, the body weights of children and youth aged ≥6 years in the 1988–1994 national survey were not used in the development of the charts [18]. This was done because the gain in weight was considered undesirable from a public health perspective and to avoid an upward shift of the percentile curves for weight and BMI. Changes in heights of American girls and boys have been relatively small across subsequent U.S. national surveys through 2014, but weights have increased over time [19,20,21]. 

Since sport is selective [1], a second preliminary analysis addressed potential size variation by competitive level. Players aged >11 years were classified as local or elite based on descriptions in the respective studies. Classifying players aged 8–10 years as elite is not warranted [22]. Age, height and weight of local and elite players were compared by two-year CA groups with analyses of covariance (ANCOVA), with year of study as the covariate. 

All data were then combined, and composite means of the mean CAs, heights and weights were calculated by single-year CA groups from 9 to 18 years; the single 8-year-old sample was included with the 9-year-olds. Mean heights and weights were plotted by CA relative to U.S. reference medians and the 25th and 75th percentiles for girls [17]. Allowing for limited numbers at the younger ages, moving averages, with a window of three measurements, of mean heights and weights were also calculated and fitted with Preece–Baines Model 1 [23,24] to estimate ages at peak height velocity (PHV) and peak weight velocity (PWV) using Dell Statistica 13 [25]. Although designed for longitudinal data, the Preece–Baines model has been applied to cross-sectional data [26]. 

### 2.3. Maturation

Skeletal age (SA) and secondary sex characteristics (breasts, pubic hair) are established indicators of maturity status—state of maturation at the time of observation. Both are often labeled as “invasive” as the former requires a small dose of radiation and the latter is often considered as intruding an adolescent’s privacy. Ages at menarche and at peak height velocity (PHV) are indicators of maturity timing—ages at which specific maturational events are attained. Both indicators of timing require longitudinal data that span adolescence [27]. Unfortunately, longitudinal data for female soccer players that span adolescence are not available.

In this context, non-invasive predictions of maturity status and timing are increasingly used in studies of youth athletes [28]. The use of current height expressed as a percentage of predicted adult height without SA was proposed as an indicator of maturity status in the 1980s [29], and sex-specific equations for the prediction of adult height based on the age, height and weight of the individual and the mid-parent height of her/his biological parents were developed [30]. The latter equations are often used in the context of bio-banding [31]. As commonly applied, the heights and weights of players are measured, while the parental heights are reported. 

Sex-specific equations for the prediction of maturity offset, defined as time before PHV [32], are also increasingly used. The equations require CA, height, weight, sitting height and estimated leg length. Modified prediction equations using age and height in girls and age and sitting height or age and height in boys are also available [33]. By definition, predicted maturity offset is an indicator of maturity timing; predicted age at PHV is estimated as the difference between CA at prediction and maturity offset.

The earliest study of maturity status (skeletal age) in female soccer players was published in 1981 [2], while an indicator of maturity was reported in 36 studies (22%) spanning 1995 to 2020. The use of established maturity indicators, however, was limited (Appendix A). Only one recent study in 2020 considered SA, while pubertal status was evaluated in nine studies spanning 2010 to 2019. The ages at menarche, with one exception, were based on the retrospective (recall) method in seven studies of adolescents and young adults spanning 1995 to 2017. Predicted maturity offset was used in 16 studies spanning 2012 to 2020, while the percentage of predicted adult height attained at the time of observation was used in three studies: 2002, 2018 and 2020. 

## 3. Results

### 3.1. Height and Weight

The mean heights of soccer players in studies spanning the years 1992–2009, 2010–2017 and 2018–2020 overlapped and spanned the 25th to 75th percentiles of the U.S. reference (Appendix A), although most means were at or above the reference medians at 15–18 years. The trend was similar for body weight, but most means in the three intervals were at or above the reference medians beginning at about 12–13 years (Appendix A). The ANOVAs indicated only one significant difference among the CAs, heights and weights of players in the three time intervals (Table 1). Players 13–14 years of age in 1992–2009 were significantly heavier than players in 2010–2017 (*p* < 0.05), but the effect size was small (η_p_^2^ = 0.07). By inference, secular changes in heights and weights of female soccer players were negligible between 1992 and 2020. 

The ages, heights and weights of players aged 13–14 and 17–18 years classified as elite or local did not differ (Table 2), while local players aged 11–12 years were significantly taller and heavier than elite players (*p* < 0.05) and elite players aged 15–16 years were significantly taller than local players (*p* < 0.05). The effect sizes (η_p_^2^), 0.22 to 0.36, suggested a moderate effect.

The mean heights of the total sample of soccer players by CA group and the estimated fit of the Preece–Baines model to the moving averages are illustrated in Figure 1. The mean heights were, on average, taller than the reference medians among players aged <11 years, approximated the medians through 14 years, and were above the medians through 18 years. The estimated age at PHV based on the Preece–Baines model was 11.55 years with a standard error of estimate (SEE) of 0.56 cm. The latter is an estimate of the goodness of fit of the model. The corresponding trends for body weight are shown in Figure 2. The weights of soccer players were consistently above the reference medians across the age range. The estimated age at PWV was 12.22 years with an SEE of 0.67 kg.

### 3.2. Maturity Status

#### 3.2.1. Skeletal Age (SA)

Two studies, separated by 40 years, considered the skeletal maturity of female soccer players. The earlier study [2] involved 23 players observed at 13 and 16 years of age and used a local adaptation [34] of the Greulich–Pyle method [35] to estimate SA. CAs and SAs of the soccer players were, on average, similar at both ages [2]. The more recent study of a large sample of Portuguese players indicated Greulich–Pyle SAs that were, on average, in advance of CAs at 12 and 13 years, equivalent at 14 years and slightly delayed at 15 and 16 years; when assessed with the Fels method [36], the SAs were advanced, on average, relative to CAs from 12 to 16 years [37]. 

#### 3.2.2. Pubertal Status

Nine studies evaluated pubertal status. The status was assessed at a clinical examination in one study [38], while self-assessments of stage of breast [39] or pubic hair [40] development or of “Tanner stages” [41,42,43,44,45] were used in seven studies. The latter did not differentiate between breasts and pubic hair. The samples spanned variable CA ranges: 8–14, 9–13, 9–18, 11–14, 12–15 and 13–15 years. Players were generally classified as pre-pubertal, pubertal or post-pubertal (though procedures and labels varied) or the distributions of stages were simply described for the sample. The variation in pubertal status by CA or variation in CA within and between pubertal groups was not addressed. It is thus difficult to ascertain the sexual maturity status of the samples relative to a reference for the general population. One study used self- and parental-assessments of pubertal status to identify pre-pubertal (9.9 ± 0.3 years) and post-pubertal (14.6 ± 0.5 years) players [46]. 

#### 3.2.3. Age at Menarche

Seven studies [47,48,49,50,51,52,53] reported ages at menarche among soccer players (Table 3). Only one study was focused on growth and maturity status per se [47] and provided estimates of age at menarche with the status quo and retrospective methods [27]. The status quo method was used with players aged 10–18 years; it required the CA of the player and whether or not she had attained menarche (yes/no). Based on probit analysis, the median age at menarche of the sample was 12.9 ± 0.9 years and was identical with the mean recalled age at menarche among adolescent and collegiate players aged 14–23 years in the study (12.9 ± 1.3 years) (Table 3). In one study [52], 48 of 75 players aged 10 through 14 years (64%) had not yet attained menarche, but the distribution of players by menarcheal status (yes/no) within single-year CA groups was not reported. A study of pubertal status in 35 Spanish players of 11 through 13 years of age (12.8 ± 0.8 years) noted that 20 players had already attained menarche but did not report the distribution by CA [43].

The ages of players in studies using the retrospective method ranged from 13 to 30 years and sample sizes were small (*n* = 13–16) in three studies. The seven mean ages varied between 12.7 and 13.0 years with standard deviations of 0.7 to 1.3 years. The studies spanned 1995 to 2017 and thus indicated no evidence of secular change.

#### 3.2.4. Percentage of Predicted Adult Height

Three studies used the percentage of predicted adult height attained at the time of observation as the indicator of maturity status. Maturity status so defined was used as a continuous variable among players aged 9–15 years and was unrelated to perceptions of adult autonomy support [54]. The method was used in two studies to classify players by pubertal status. Among 34 players (13.3 ± 1.5 years), nine were classified as pubertal (88–94%) and 25 as post-pubertal (>94%); the ages of players in each group were not indicated [55]. In the other study, players aged 9–17 years were classified as pre-pubertal (<87%, *n* = 17, 10.3 ± 0.6 years), pubertal (87–94%, *n* = 32, 11.9 ± 0.8 years) or post-pubertal (>94%, *n* = 90, 14.6 ± 1.6 years) [56].

#### 3.2.5. Predicted Maturity Offset/Age at PHV

Predicted maturity offset was used in 16 studies spanning the years 2012–2020. The original sex-specific equation [32] was used in all but one study (Appendix A). The ages of players at prediction ranged from 8 to 18 years, but the mean CAs largely spanned 11–15 years. Most studies reported predicted maturity offset, while four reported predicted age at PHV; one study reported both. Several studies compared the characteristics of players classified as pre-, at- or post-PHV (i.e., maturity status); the results for predicted offset or age at PHV were largely descriptive.

The means for predicted maturity offset in soccer players increased linearly with mean CA (Figure 3A) and mean height (Figure 3B) at prediction. The mean predicted ages at PHV are shown in Figure 4, which also includes corresponding data for girls in the Fels Longitudinal Study for comparison [57]. The predicted ages at PHV in soccer players overlapped those for the Fels sample, tended to increase with CA at prediction and were later than the observed age at PHV in the Fels sample beginning at about 10 years of age. 

## 4. Discussion

### 4.1. Height and Weight

The heights and weights of female soccer players showed no secular variation between 1992 and 2020 (Table 1). Although not directly comparable, male soccer players in 121 studies spanning the years 2000–2015 were taller and heavier than players in 23 studies spanning the years 1978–1999 [58]. 

Female players aged <11 years were, on average, taller and heavier than the U.S. reference, although data were limited. The heights of the players approximated the reference medians through 14 years and were above the medians and approached the 75th percentiles in late adolescence (Figure 1). Body weights, in contrast, were above the reference medians throughout the age range (Figure 2). The trends suggested the persistence and perhaps selection of taller and heavier players during adolescence. The mean heights (158.0 to 169.3 cm) and weights (54.5 to 66.1 kg) of late adolescent players aged 17–18 years overlapped the mean heights (158.1 to 171.0 cm) and weights (55.2 to 65.7 kg) of adult soccer players aged 19–27 years surveyed between 1986 and 2020 (Appendix A). 

The tallness in association with a heavier mass among late adolescent players probably reflected a more muscular physique and systematic training. Training programs aimed at the development of strength and power often enhance lean tissue mass [59,60], while regular intensive training does not influence growth in height or the timing of the adolescent spurt [61]. Estimates of relative fatness among late adolescent soccer players were variable, likely reflecting variation among samples per se and in methodology—predictions from skinfolds, hydrostatic weighing, bioelectrical impedance and DEXA [12,13,60]. 

The estimated age at PHV based on modeling moving averages for the soccer players was 11.55 years. Allowing for limitations associated with modeling cross-sectional data [26], the estimate was within the range of mean ages at PHV in longitudinal studies of European and North American girls [27,62]. The age at PHV occurs earlier, on average, among Japanese girls [27], although soccer players from Japan accounted for only 12 data points (5%) spanning 12–18 years of age. The estimated age at PHV for soccer players was somewhat earlier than the ages at PHV in longitudinal samples of youth of European ancestry participating in athletics, rowing and mixed sports, while ages at PHV for Japanese girls active in several sports, though not soccer, also tended to be earlier than the mean ages for athletes of European ancestry [63]. Consistent with observations in longitudinal samples, the estimated age at PWV of female soccer players (12.22 years) occurred, on average, after PHV, but was at the lower end of the range of mean ages noted in longitudinal samples of European and North American girls [27].

The Preece–Baines model also provided an estimate of height at PHV (149.6 cm) and young adult height (165.6 cm). The estimated height at PHV in soccer players was 90.3% of adult stature, which was similar to estimates in longitudinal studies of girls in Europe, 90.8% [64] and 91.5% [65], and the U.S., 90.0% and 90.5% [66]. Overall, the estimated age at PHV and height at PHV as a percentage of adult height among female soccer players approximated the estimates for girls in the general population who were “on time” or average in these biological landmarks.

### 4.2. Skeletal Maturation

Only two studies considered the skeletal maturity status of female soccer players. The earlier study observed no differences between the CA and SA of players at 13 and 16 years of age [2], while the more recent study noted variation between the Greulich–Pyle [35] and Fels [36] SAs and a tendency for soccer players to be somewhat advanced in SA relative to CA with the Fels method [37]. Comparative skeletal maturation data for female athletes in team sports are limited to the earlier study [2], which noted no differences in the CA and SA of volleyball players at 14 and 17 years of age. SA data are more available for female athletes in several individual sports [67]. Based on the difference between SA and CA, late-maturing girls were predominant during adolescence among artistic gymnasts. Although the majority of swimmers < 14 years of age tended to be average in SA, equal numbers of swimmers aged 14–15 years were average or advanced in SA and the majority at 16–17 years of age were skeletally mature. Among track and field athletes, in contrast, maturity status varied by discipline [67], while the SAs of elite female tennis players were generally similar to the CAs among U12 players but advanced relative to the CAs among U14 and U16 players [68].

### 4.3. Pubertal Status

Allowing for methodological inconsistencies, e.g., specifically combining stages of breast and pubic hair or not specifying whether stage of breast or pubic hair was used, the studies of soccer players did not address sexual maturity status per se. Players were classified as pre-, peri- and/or post-pubertal or distributions of stages were simply described without considering variation in CA. Of relevance, Stage 1 of breast or pubic hair development is the pre-pubertal state, while Stage 2 of each characteristic marks the overt manifestation of puberty. Stages 2, 3 and 4 mark the pubertal state, while Stage 5 of breast or pubic hair indicates the post-pubertal or mature state [27]. It is erroneous to classify players who have attained Stage 2 as pre-pubertal or players who have attained Stage 4 as post-pubertal [38,43,44].

### 4.4. Age at Menarche 

The single estimate for youth soccer players based on the status quo method (12.9 years) was similar to the recalled ages at menarche (Table 3). The estimate was also consistent with estimates for youth athletes in several sports, including athletics, swimming and several team sports, while estimated ages at menarche for youth divers, figure skaters and artistic gymnasts were, on average, later [63]. 

The retrospective method requires the athlete to recall her CA at menarche. It relies on memory and may be influenced by recall bias and error, while some individuals recall only the whole year, presumably age at last birthday. Nevertheless and allowing for small samples in three studies, mean recalled ages at menarche of soccer players spanned a narrow range (12.7 to 13.0 years) and indicated no secular effect. The means were also within the range of those for the general population [69,70,71,72,73,74] and were at the low end of recalled ages at menarche for athletes in team sports [27]. In contrast, a study of adult soccer players from Kosovo reported a mean recalled age at menarche of 13.5 ± 1.3 years [75]. Age at menarche was presumably obtained by questionnaire as one question asked, “Did you start exercising before menarche?” The mean for soccer players was identical to the mean ages at menarche among handball players (13.5 ± 1.0 years) and non-athletes (13.5 ± 1.0 years) [75].

Although the mean age at menarche in one sample of U.S. soccer players was 13.0 ± 1.0 years [52] and consistent with other estimates (Table 3), the authors noted: “This is the first study to describe a delay in the onset of menarche in a cohort of soccer athletes” (p 3). “Delay” implies that something associated with the sport influenced age at menarche. Although training is often indicated as a factor contributing to later ages at menarche in some athletes, the general consensus is that training per se does not affect menarche [76]. Training, moreover, is only one aspect of the sport environment. Other factors include factors associated with selection, persistence and exclusion (cutting, dropout) and behaviors of coaches and trainers. The latter often place undue pressures upon young athletes in the context of diet and weight control, and extreme dietary restriction may delay menarche [76]. Age at menarche is also a heritable characteristic as evident in studies of twins [27], mother–daughter and sister–sister similarities in the general population and in athletes [77] and ethnic variation [78,79].

### 4.5. Percentage of Predicted Adult Height

The percentage of predicted adult height attained at the time of observation is increasingly used in studies of bio-banding among youth athletes, more so in males than in females [31]. Data relating maturity status based on predicted adult height to SA are lacking for female soccer players, but the indicator had moderate concordance with SA in female tennis players [80]. 

The percentage of predicted adult height attained at the time of observation was used in two studies to classify players spanning a relatively broad CA range by pubertal status. Criteria for classifying youth players as pre-pubertal, pubertal or post-pubertal should be evaluated relative to observations from longitudinal studies. For example, among girls in the Zurich Longitudinal Study, estimated CA and percentage of adult height *on entry into* stage 2 of pubic hair (PH2) were 10.7 ± 1.1 years and 85.8 ± 2.9%, respectively [81]. In one study of soccer players [56], a percentage of predicted adult height of <87% defined the pre-pubertal state. Girls so classified were aged 10.3 ± 0.6 years with a percentage adult height of 84.0 ± 1.8%. Given the Zurich data, it is possible that some of the girls with a percentage of predicted adult height of <87% were in fact pubertal. Similarly, using a percentage of predicted adult height of >94% to indicate the post-pubertal state [55,56] merits evaluation. Among girls in the Zurich study, estimated CA and percentage of adult height *on entry into* stage 4 of pubic hair (PH 4), which is still the pubertal state, were 12.7 ± 1.2 years and 94.2 ± 2.1%, respectively, while estimated CA and percentage of adult height on entry into PH 5, the mature state, in Zurich girls were 13.7 ± 1.2 years and 97.0 ± 1.5%, respectively [81]. 

### 4.6. Predicted Maturity Offset

Predicted maturity offset and/or predicted age at PHV (CA minus offset) was generally accepted as a maturity indicator or was used to classify players as pre-, at- or post-PHV without considering variation in CA per se. Predicted maturity offset increased linearly with CA (Figure 3A) and with height (Figure 3B) at prediction in the samples of soccer players. The trends in soccer players were consistent with CA-related trends in three longitudinal studies of girls [57,82,83], which also noted reduced variability in predicted maturity offset and in ages at PHV, and major limitations of the predictions among early- and late-maturing girls defined by observed age at PHV. 

The limitations of predicted maturity offset as a valid indicator of the time before or after PHV should be noted. The interval of PHV is central to the long-term athlete development program for youth players in soccer [15] and other sports [84,85]. Awareness of the potential for misclassifications associated with predicted maturity offset and implications for player development require attention. 

## 5. Summary

Relatively few of the studies considered in this narrative review (6%) were specifically focused on the growth and maturation of female youth soccer players. Of the studies reporting heights and weights, most (40%) were focused on changes in fitness and performance with CA and on training and match play, while many focused on the biomechanics and kinematics of the knee and hip, largely in the context of jumping (16%) and injuries and heading (16%). Other studies focused on bone health and body composition (10%), psychological variables (6%), nutritional status (4%) and general physiology (3%). Only one of seven studies reporting ages at menarche in soccer players was focused on growth and maturation per se [47], five were set in the context of bone health [48,49,50,51,53] and one considered body perceptions [52]. 

Allowing for the variety of studies, the heights and weights of players showed negligible secular variation between 1992 and 2020. Though limited to few observations, players aged <11 years were, on average, taller and heavier than the reference. Subsequently, the heights approximated the reference median through 14 years and approached the 75th percentile in late adolescence, while the weights were above the reference median from 12 through 18 years of age. The trends suggested the persistence of and/or selection for taller and heavier players during the adolescent years. 

The estimated age at PHV based on the Preece–Baines model applied to the moving averages for height was 11.55 years and the mean ages at menarche ranged between 12.7 and 13.0 years. Both estimates of maturity timing among soccer players were within the range of mean ages at PHV and menarche in European and North American samples. Data for skeletal and sexual maturity status and for percentage of predicted adult height among female players are limited, while predicted maturity offset and age at PHV are increasingly used as a maturity indicator.

## 6. Implications 

Individual differences in growth and maturation play a central role in the development of youth athletes, and the interval of PHV is central to the long-term development model proposed for female soccer players ([15], p. 11): 


*“Optimal aerobic training begins with the onset of Peak Height Velocity (PHV), more commonly known as the adolescent growth spurt. Strength training has two windows of accelerated adaptation in this phase. Window 1 is immediately after PHV and window 2 begins with the onset of menarche.”*


PHV specifically refers to the estimated maximal rate of growth in height during the adolescent spurt, while menarche occurs, on average, after PHV. Longitudinal data that span adolescence, 8 through 16–17 years of age in girls, are required to estimate PHV and age at PHV. Unfortunately, longitudinal data spanning the adolescent years are not available for female soccer players.

The adolescent growth spurt begins with an acceleration in the rate of growth in height during middle or late childhood (take-off); the rate of growth increases until it reaches a maximum (PHV), then decelerates and eventually ceases in late adolescence [27]. Allowing for different methods of estimating parameters of the growth spurt, descriptive statistics and ranges of variability for ages at take-off of the spurt and in ages at PHV in three longitudinal studies of European girls are summarized in Table 4. The means are reasonably similar, but the ranges of variation are considerable. Corresponding statistics for the interval between age at take-off and age at PHV in the Polish longitudinal sample were 3.1 ± 0.8 years with a range of 0.5 to 5.4 years (calculated from data reported in reference [82]).

Discussions of the need to monitor the adolescent spurt in the context of models for athlete development do not ordinarily address the range of individual differences in parameters of the growth spurt or the need to begin monitoring the growth of female athletes longitudinally at relatively young ages. The developmental model for female soccer [15] does not specify the method for identifying the onset of the growth spurt or for estimating age at PHV, while the Long-Term Athlete Development (LTAD) model [85,86] suggests monitoring growth in height over short intervals to estimate growth velocities and to monitor sudden change in velocity that may be indicative of the growth spurt. Estimated increments in growth over short intervals (3–4 months) must be interpreted with care. They should be adjusted for the interval between measurements and should consider measurement error at each observation (both inter- and intra-observer variability). Height measurements also vary with time of day (diurnal variation), tend to be reduced after intensive physical activity and also vary with the season of the year [27]. 

Although predicted maturity offset (defined as the time before PHV) and estimated age at PHV (CA at prediction minus maturity offset) are increasingly applied in studies of youth athletes, including female soccer players (Figure 3 and Figure 4), intra-individual variation depending on CA at prediction is considerable. The predictions also have major limitations differentiating between early- and late-maturing youth defined by observed age at PHV [75,82,83]. Predicted ages at PHV are later than observed age at PHV among early-maturing girls and earlier than observed age at PHV in late-maturing girls. Practitioners using predicted maturity offset per se or variations of the method to identify when players enter and exit the interval of the adolescent growth spurt should be aware of these limitations and employ the methods with caution. 

Predicted maturity offset also has major limitations when used to classify players by maturity status or to adjust fitness and performance scores to accommodate individual differences in maturation. If predicted offset is used to inform training design and prescription, it is essential that variation in CA and height at prediction and error associated with the prediction equations be considered. Perhaps additional or alternative methods might be considered as a complement, e.g., percentage of predicted adult stature attained at the time of observation. These and other issues related to the growth and maturity status and timing of youth athletes are discussed in more detail in several papers cited in this review [31,63,87].

## Figures and Tables

**Figure 1 ijerph-18-01448-f001:**
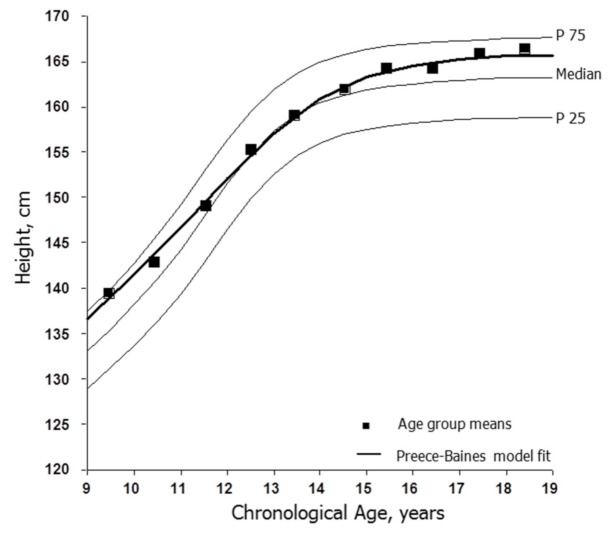
Age-group means for heights of female soccer players (■) and the fit of the Preece–Baines Model 1 (solid line) applied to the moving averages of mean heights plotted relative to reference medians and the 25th and 75th percentiles for U.S. girls [17].

**Figure 2 ijerph-18-01448-f002:**
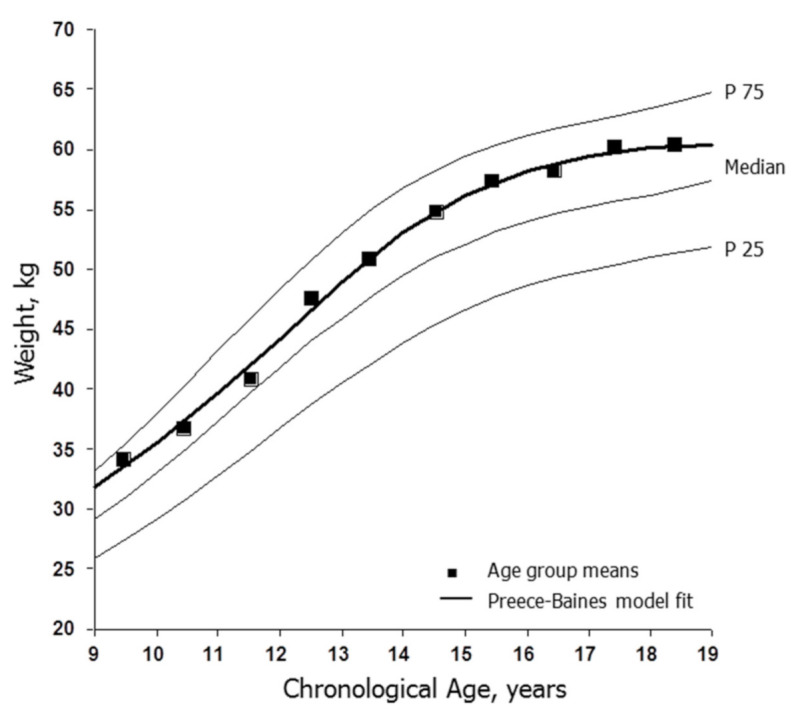
Age-group means for weights of female soccer players (■) and the fit of the Preece–Baines Model 1 (solid line) applied to the moving averages of mean heights plotted relative to reference medians and the 25th and 75th percentiles for U.S. girls [17].

**Figure 3 ijerph-18-01448-f003:**
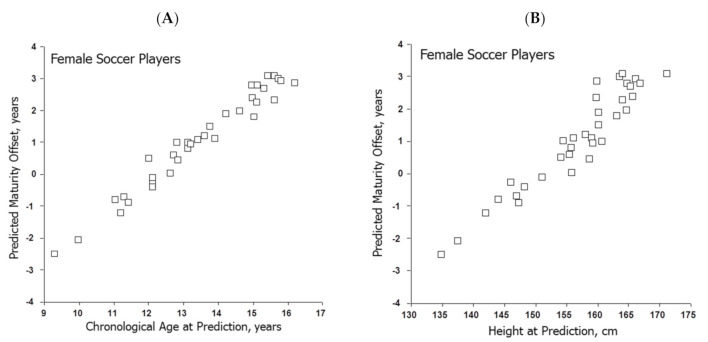
Means for predicted maturity offset (MO) in samples of female soccer players plotted relative to (**A**) mean chronological age and (**B**) mean height. References for the studies of soccer players are indicated in Appendix A: 2010–2017: [60–65]; 2018–2020: [58–68].

**Figure 4 ijerph-18-01448-f004:**
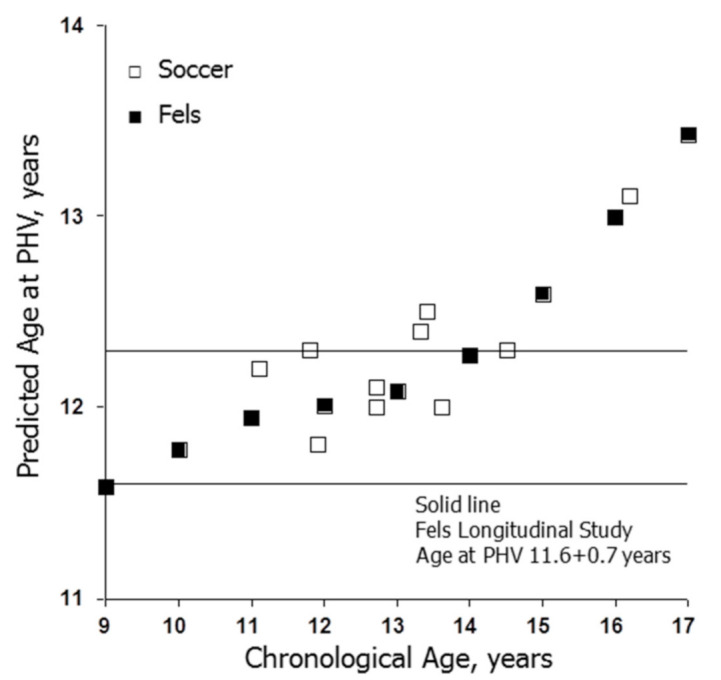
Mean predicted ages at peak height velocity (PHV) in girls aged 9 through 17 years from the Fels Longitudinal Study [57] and for female soccer players. The solid lines indicate the mean plus one standard deviation for observed age at PHV in the Fels Longitudinal Study. References for studies of soccer players are indicated in Appendix A: 2010–2017: [60]; 2018–2020: [59,64,66]; the means were calculated for select and non-select players combined in each CA group for reference [64].

**Table 1 ijerph-18-01448-t001:** Number of data points (N) ^†^, means (M) and standard deviations (SD) based on the means for chronological age (CA), height and weight of female soccer players by two-year age groups in studies spanning three intervals, 1992 to 2009, 2010 to 2017 and 2018 to 2020, and results of ANOVAs and effect sizes (η_p_^2^).

Intervals of Studies												
Age	1992–2009 (a)			2010–2017 (b)			2018–2020 (c)					Significant
Groups	N	M	SD	N	M	SD	N	M	SD	F	η_p_^2^	Differences
CA, years												
<11	5	10.1	0.3	9	9.9	0.8	6	10.4	0.3	1.62	0.16	
11–12	7	12.0	0.4	17	12.1	0.5	29	12.2	0.6	0.51	0.02	
13–14	11	14.2	0.6	38	14.1	0.5	47	13.9	0.7	0.97	0.02	
15–16	18	15.8	0.6	35	15.8	0.6	42	15.9	0.6	0.58	0.01	
17–18	8	17.8	0.4	15	17.9	0.6	6	17.8	0.7	0.04	<0.01	
HEIGHT, cm												
<11		142.8	3.6		140.1	3.4		143.2	4.5	1.51	0.15	
11–12		153.3	2.1		152.2	3.7		153.4	5.5	0.36	0.01	
13–14		162.0	3.2		160.5	3.4		160.5	2.5	1.30	0.03	
15–16		164.0	2.1		164.2	2.1		164.6	2.5	0.31	0.01	
17–18		164.9	2.5		166.4	2.8		165.1	3.6	0.87	0.06	
WEIGHT, kg												
<11		36.9	1.5		35.1	3.0		36.4	3.1	0.77	0.08	
11–12		46.1	3.8		44.2	4.8		45.3	5.5	0.39	0.01	
13–14		55.6	3.4		52.5	3.7		52.9	3.5	3.44 *	0.07	a > b
15–16		57.5	2.2		57.3	2.6		58.1	2.6	0.92	0.02	
17–18		60.5	2.5		60.5	2.9		59.6	3.2	0.24	0.02	

^†^ data points are the means for mean ages, heights and weights reported in the available studies or calculated from data available for six studies; * *p* < 0.05.

**Table 2 ijerph-18-01448-t002:** Number of data points (N), means (M) and standard deviations (SD) based on means for the age, height and weight of female soccer players classified as local or elite in four chronological age groups, results of age-group-specific ANCOVAs controlling for the year of study, and effect sizes.

Age Groups	Local			Elite				
	N	M	SD	N	M	SD	F	η_p_^2^
AGE, years								
11–12	32	12.2	0.5	21	12.0	0.6	2.31	0.04
13–14	56	14.1	0.6	40	14.0	0.6	0.96	0.03
15–16	40	15.8	0.6	55	15.8	0.6	0.01	<0.01
17–18	6	18.0	0.6	23	17.8	0.6	0.37	0.01
HEIGHT, cm								
11–12		154.1	3.5		151.2	5.6	6.17 *	0.11
13–14		160.6	3.0		160.9	2.9	3.22	0.03
15–16		163.7	2.0		164.7	2.4	4.33 *	0.05
17–18		164.8	3.4		166.0	2.8	2.40	0.08
WEIGHT, kg								
11–12		46.4	4.7		42.8	4.9	7.27 *	0.13
13–14		53.4	3.9		52.5	3.2	2.49	0.03
15–16		57.3	1.8		58.0	2.9	1.29	0.01
17–18		60.5	3.5		60.3	2.7	1.36	0.05

* *p* < 0.05.

**Table 3 ijerph-18-01448-t003:** Descriptive statistics for chronological age and age at menarche (mean (M) or median (Md) and standard deviation (SD)) in female soccer players; studies are listed by year of publication.

							Age at Menarche, Years			
Year,				Chronological Age, Years			Recall		Status Quo	
Reference	Country	Method	N	Range	M	SD	M	SD	Md	SD
1995 [47]	US	Probit	82	10–18					12.9	0.9
		Recall	62	14–23	17.1	2.3	12.9	1.3		
1996 [48]	Sweden	Recall	96	13–28	18.3	4.0	12.8	1.3		
1997 [49]	US	Recall	16		15.1	0.7	12.9	1.2		
2000 [50]	Sweden	Recall	15		17.6	0.8	12.7	0.7		
2000 [51]	Sweden	Recall	51	14–19	16.3	1.4	13.0	0.9		
2016 [52]	US	Recall	145	15–30			13.0	1.0		
2017 [53]	Sweden ^1^	Recall	13	13–16	15.3	0.7	13.0	0.8		

^1^ The initial sample was 19 players; only 15 completed the questionnaires and, of the 15, ages at menarche were available for 13 players.

**Table 4 ijerph-18-01448-t004:** Means (M), standard deviations (SD) and ranges for ages at take-off (initiation) of the growth spurt and at PHV in three longitudinal series of European girls.

		Age at Take-Off, Years			Age at PHV, Years		
		M	SD	Range	M	SD	Range
Swiss [86]	110	9.6	1.1	6.6–12.9	12.2	1.0	9.3–15.0
British [23]	23	9.0	0.7	7.7–10.0	11.9	0.7	10.3–13.2
Polish [82]	198	8.9	1.1	6.3–12.0	11.9	1.0	9.0–14.8

## Data Availability

The data are available in the respective studies enumerated in Appendix A.

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
