# Peer review of "Growth and Maturity Status of Female Soccer Players: A Narrative Review"

_ijerph, 2021, doi:10.3390/ijerph18041448_

Round 1
Reviewer 1 Report
General Comments:
The study Growth and Maturity Status of Female Soccer 2Players: An Overview is an interesting paper and could be very informative to the readers of International Journal of Environmental Research and Public Health. Thus, I would encourage the authors to polish the study.
The introduction gives the reader a good overview of the currently published reviews. It manages to work out the limitations of the reviews on growth and maturity status of female soccer players and thus leads stringently to the research problem and the research question.
In the Material and Methods chapter, the search strategy needs to be presented more transparently to the reader. The number of individuals who completed the title and abstract screening is missing, a calculation of rater-reliability, if necessary, and the inclusion and exclusion criteria need to be presented in more detail. Bias analyses (possibly funnel plots, Eggers regression or trim and fill) would better classify the results of the selected studies. The authors themselves already point out potential problems in lines 95 to 97. Therefore, such procedures could be helpful for the investigation.
The presentation of results was well implemented. However, a flow chart should be added to the Results chapter so that the process of inclusion and exclusion of studies can be better understood by the reader. This would also help to shorten chapter 2 (please also see specific comments)
The discussion is structured, but unfortunately, from the reviewer's point of view, the significance and relevance of the data for the training practice is clearly too short. This is also reflected in very generalized statements in the sentence in lines 381 to 383 and 390 to 393. The data should therefore also be discussed against the background of training practice and possible consequences.
Specific comments:
Line 50 and 51: please check if the sentence in line 50 and 51 is relevant in terms of working out the research question. Please delete the sentence if necessary.
Line 65 to 83: this section needs to be shortened
Line 94: Please explain briefly in a supplementary sentence why the defined age categories were chosen.
Reviewer 2 Report
A clearly written manuscript that describes some interesting data, although in a niche area that will be of interest to a relatively small readership.
A few points that should be addressed.
Page 2, line 58. More details of the literature surveyed (databases, via PubMed etc.) are required. How was the survey conducted, did it follow establish standards for literature review?
Materials and methods. Much of this reads like a results section or background information. Th authors might consider limiting methods to the search strategy and the analysis and place much of the information here in an introductory section to the Results.
Page 3, line 114. "Moving averages" - over how many data points?
Page 4, line 148. Table 1 should be near here. At present it does not appear until the Discussion on page 8.
Page 7, Fig 3. I wonder if best-fit trend lines are warranted?
Round 2
Reviewer 1 Report
The authors have revised their manuscript and in my opinion the article has been improved. Thank you for that. The clarification of the scientific strategy (narrative review) has contributed to the improvement of the manuscript. The approach of the authors becomes clearer for the reader.
However, there are still some passages that need to be revised:
Line 50 and 51, additional Comment. I totally agree with you that “growth and maturation …may influence change of direction speed tests and other functional indicators”. However, you also name other functional indicators yourself in your response. If you want to clarify the influence of growth and maturity on performance determinates, which seems not necessary at this (not leading to the research question), please open the wording for other “other functional indicators”.
Line 79 to 91: section needs to be shortened or if you want to draw attention to this issue, then integrate this information into the introduction. This could also lead to a further strengthening of the research question.
Line 104-106: Thank you very much for elaborating on my request. This would need to be integrated into the manuscript for clarification.
Training practice and possible consequences and Line 443 to 447: I can certainly understand your comments, which in particular point out limitations of current research. With regard to the consequences, I was asked for an outlook e.g. on a sensible structuring (especially against the background of the currently predominantly practiced procedures: chronological division of training groups) of junior sports, taking into account the limitations mentioned. It is at this point that a narrative review opens up possibilities to show an outlook based on the expertise of the authors. Especially coaches in junior sports could benefit from this with regard to the composition of training groups.
